# Effects of Dietary Fiber on Growth Performance, Nutrient Digestibility and Intestinal Health in Different Pig Breeds

**DOI:** 10.3390/ani12233298

**Published:** 2022-11-25

**Authors:** Jiahao Liu, Yuheng Luo, Xiangfeng Kong, Bing Yu, Ping Zheng, Zhiqing Huang, Xiangbing Mao, Jie Yu, Junqiu Luo, Hui Yan, Jun He

**Affiliations:** 1Institute of Animal Nutrition, Sichuan Agricultural University, Chengdu 610000, China; 2Key Laboratory of Animal Disease-Resistant Nutrition, Chengdu 610000, China; 3Institute of Subtropical Agriculture, Chinese Academy of Sciences, Changsha 410000, China

**Keywords:** dietary fiber, nutrition, intestinal barrier function, volatile fatty acid, pigs

## Abstract

**Simple Summary:**

Dietary fiber has been long been established as a nutritionally important, health-promoting food ingredient. Our aim was to explore differences in physiological and metabolic adaptations to DF in two pig breeds, and the mechanisms behind DF-regulated intestinal health have also been investigated. Our results showed a difference in dietary fiber utilization by the two pig breeds (Taoyuan and Duroc), which might be connected with the integrity of the intestinal epithelium and the microbial activity of the gastrointestinal tract, and the results may also suggest a beneficial role of dietary fiber in regulating intestinal health.

**Abstract:**

To explore the effect of dietary fiber on growth performance and intestinal health in different pig breeds, forty Taoyuan and Duroc pigs (pure breeds) of 60 days of age were randomly divided into a 2 (diet) × 2 (breed) factorial experiment (*n* = 10), and fed with a basal diet (BD) or high-fiber diet (HFD). The trial lasted for 28 d, and results showed that the Taoyuan pigs had a higher average daily feed intake (ADFI) than the Duroc pigs (*p* < 0.05). The average daily gain (ADG) and digestibilities of gross energy (GE) and crude protein (CP) were higher in Taoyuan pigs than in the Duroc pigs under HFD feeding (*p* < 0.05). The HFD increased the superoxide dismutase (SOD) and catalase (CAT) activity in Taoyuan pigs (*p* < 0.05). Interestingly, Taoyuan pigs had a higher jejunal villus height and ratio of villus height to crypt depth (V/C) than the Duroc pigs. The HFD significantly improved the villus height and V/C ratio in duodenum and jejunum (*p* < 0.05). The HFD also increased the jejunal maltase and ileal sucrase activities in Duroc and Taoyuan pigs, respectively (*p* < 0.05). Taoyuan pigs had a higher expression level of duodenal fatty acid transport protein-1 (FATP-1) than the Duroc pigs (*p* < 0.05). Furthermore, the HFD acutely improved the expression levels of ileal SGLT-1 and GLUT-2, and the expression levels of jejunal occludin and claudin-1 in Taoyuan pigs (*p* < 0.05). Importantly, Taoyuan pigs had a higher colonic *Bifidobacterium* abundance than the Duroc pigs (*p* < 0.05). The HFD not only elevated the colonic *Lactobacillus* abundance and butyrate acid content in Taoyuan pigs, but also increased the acetic and propionic acid contents in both the pig breeds (*p* < 0.05). These results indicated a difference in dietary fiber (DF) utilization by the two pig breeds, and results may also suggest a beneficial character of DF in regulating intestinal health.

## 1. Introduction

Dietary fiber (DF), usually defined as the indigestible portion of plant-derived foods [1], has been long been established as a nutritionally important, health-promoting food ingredient [1,2]. However, a dominant concern for mono-gastric animals (e.g., pigs) is that a high-fiber diet (HFD) is associated with reduced nutrient utilization and low net energy values, as the DF cannot be digested by endogenous digestive enzymes [3,4]. Currently, a number of studies have indicated a positive role of DF in maintaining regular physiological functions of the digestive tract, and a significant reduction in DF consumption is always linked to an increased prevalence of gut diseases such as inflammatory bowel disease [5,6]. It is a well-known fact that the influence of DFs on animals depends on their levels, types, and physicochemical properties [7].

DF components such as cellulose, hemicellulose, and pectin are the main components of plant cell walls and cannot be broken down by mammalian enzymes, but can be fermented through a wide variety of microorganisms in the hindgut [8]. In contrast, lignin, a high molecular weight polymer, cannot be broken down in the digestive tract [9,10]. Previous studies indicated that various microorganisms in the hindgut can ferment DF to produce a number of short-chain fatty acids (SCFAs), which not only promote the growth of beneficial microorganisms, but also improve the intestinal integrity and functions [11,12]. A diet rich in DF has also been found to reduce the risk of many dietary problems such as cardiovascular disease, type 2 diabetes, and Crohn’s disease in humans [13,14,15]. Moreover, a recent study indicated that DF supplementation has no effect on the growth of growing-finishing pigs, but can obviously improve their meat quality [16].

China is the world’s largest consumer of pork, not only accounting for more than half of the global pork consumption each year, but also having a variety of pig breeds [17]. The meat produced by Chinese local pigs has been characterized by tender flesh, high intramuscular fat (IMF), homogeneous marbling, and better flavor juiciness [18,19,20]. Importantly, the Chinese local pigs can digest roughage more efficiently than other commercial pig breeds [21]. Taoyuan pigs are one of the famous local breeds in China [22]. The breed possesses all the merits of Chinese local pigs, and has been looked upon as the representative breed of the local pigs [23,24]. Duroc pigs are one of the most utilized commercial breeds because of their fast growth and high feed utilization [25,26]. This study investigated the effect of DF on the growth performance, nutrient digestibility, and intestinal health in the two pig breeds. Our aim was to explore their differences in physiological and metabolic adaptations to DF, and the mechanisms behind DF-regulated intestinal health have also been investigated.

## 2. Materials and Methods

The animal experiment in this study was carried out after approval by the Animal Care and Use Committee of Sichuan Agricultural University (Chengdu, China). The wheat bran fiber (WBF) used in this study was bought from Chengdu Tubaite Technology Co. Ltd. The total dietary fiber content in the raw material is more than 95%, and the IDF content of the wheat bran fiber raw material is 98%.

### 2.1. Experimental Design, Diet, and Animal Housing

Forty Taoyuan (average weight 13.87 ± 0.58 kg) and Duroc (average weight 18.50 ± 1.09 kg) pigs (pure breeds) of 60 days of age were randomly divided into a 2 (diet) × 2 (breed) factorial experiment (*n* = 10), exposed to a normal basal diet (BD, 3.14% dietary fiber) or high-fiber diet (HFD, 6.86% dietary fiber). The trial lasted for 28 days. The diets (Table 1) were formulated based on National Research Council 2012 (NRC, 2012) [27]. The WBF was utilized to modify the DF level in the diets. Pigs were raised individually in metabolic cages (0.7 m × 1.5 m) under an appropriate temperature (24–29 °C) and humidity (55–65%) with ad libitum access to water and food.

### 2.2. Sample Collection

The fecal samples were collected on days 26–28 of the trial. Immediately after defecation, fresh feces of each pen were collected into their own valve bags, and 10 mL of 10% H_2_SO_4_ solution was evenly added to 0.1 kg of feces to fix fecal nitrogen. On the morning of day 29, blood samples were collected through vein puncture and inject into 20 mL plain tubes. Then, these blood samples were centrifuged at 3500× *g* at 4 °C (15 min). After subsequent centrifugation, the serums were stored at −20 °C until the serum index analysis. After the blood sampling, pigs were slaughtered by electrical stunning to collect the remaining samples. The segments of the small intestine (approximately 4 cm each) were immediately separated, slowly rinsed with cold phosphate buffer, and fixed in paraformaldehyde solution for further intestinal morphological analysis. Moreover, the mucosa samples were preserved at −80 °C for convenience of examination, which were collected by scraping fragments from the small intestine by a scalpel blade.

### 2.3. Growth Performance Evaluation

The initial body weight, final body weight, and the feed intake of each pig were measured. The feed efficiency (F:G) of every pig was computed based on the average daily gain (ADG) and average daily feed intake (ADFI).

### 2.4. Apparent Total Tract Nutrient Digestibility Analysis

Diet and fecal samples were thawed, homogenized, and freeze-dried for the nutrient digestibility analysis, and the Cr_2_O_3_ served as an external indicator. The DM, CP, EE, CF, and ash contents were measured by an AOAC standard [28]. An adiabatic bomb calorimeter was utilized to measure GE. The apparent digestibility of nutrients was computed by the equation below:(1)$$\mathrm{Apparent}\;\mathrm{digestibility}\;\mathrm{of}\;a\;\mathrm{nutrient}\;\left(\%\right)=100-100*\frac{\;\mathrm{diet}\;\mathrm{Cr}2O3*\mathrm{digesta}\;\mathrm{nutrient}\;}{\;\mathrm{digesta}\;\mathrm{Cr}2O3*\mathrm{diet}\;\mathrm{nutrient}\;}.$$

### 2.5. Serum Parameter Analysis

The D-lactate (Porcine D-lactate ELISA Kit MM33732O1) and diamine oxidase (Porcine DAO ELISA Kit MM-0438O1) were measured by using enzyme-linked immunosorbent assay kits (Jiangsu Enzyme-linked Biotechnology Co., Ltd. Nanjing, China). The commercial kits purchased from Nanjing Jiancheng Biotechnology Co., Ltd (Nanjing, China) were utilized for measurement of serum parameters such as the catalase (CAT), malondialdehyde (MDA), glutathione (GSH), total superoxide dismutase (T-SOD), total antioxidant capacity (T-AOC), urea nitrogen (BUN), glucose (GLU), triglyceride (TG), and total cholesterol (TC). All procedures were carried out in strict accordance with the instructions. The specific information of these kits is as follows:

CAT (Cat. No. A007-1-1), MDA (Cat. No. A003-1-2), GSH (Cat. No. A005-1-2), T-SOD (Cat. No. A001-1-2), T-AOC (Cat. No. A015-1-2), BUN (Cat. No. C013-2-1), GLU (Cat. No. F006-1-1), TG (Cat. No. A110-1-1), TC (Cat. No. A111-1-1).

### 2.6. Intestinal Morphological Analysis

Intestinal segments fixed with 4% paraformaldehyde were dewaxed by graded anhydrous ethanol. A cross-section of every sample was prepared and dyed with hematoxylin and eosin (H&E), followed by being sealed by a neutral resin size. ImageJ software was used to measure the crypt depth and villus height of the intestine, and the ratio of villus height to crypt depth (V/C) was calculated. Ten crypt depths and villus heights were calculated and the average value was calculated [29].

### 2.7. Enzyme Activity

The intestinal mucosa was homogenized with cold saline, and then the supernatants were isolated (centrifugation at 3500× *g* for 15 min) and utilized to determine the enzyme activities of lactase, sucrase, and maltase. The measurements were carried out by using specific assay kits: sucrase (A082-2-1), lactase (A082-1-1), and maltase (A082-3-1), purchased from Nanjing Jiancheng Biotechnology Co., Ltd. (Nanjing, China).

### 2.8. Colonic Microbiological Analysis

An estimated 200 mg colon digesta was processed to obtain total DNA with the Omega Bio-Tek Stool DNA Kits for quantification real-time PCR, which was performed through the Quant Studio 6 Flex real-time PCR system (Bio-Rad). Total bacteria number was detected by the reaction which runs in a total volume of 25 μL, including SYBR Premix Ex Taq (2 × concentrated), forward and reverse primers (100 nM), DNA, RNase-free ddH2O, and 50 × ROX Reference Dye*3. All procedures of microbial real-time quantitative PCR were based on the methods reported by Wu et al. (2018) [7]. The SuperReal PreMix (Probe) kits obtained from Tiangen Biotech Co., Ltd. (Beijing, China) were utilized to determine E. coli, Bifidobacterium, Lactobacillus, and Bacillus. Each reaction was run in a total volume of 25 μL, including DNA, forward and reverse primers (100 nmol/L), 2 × Super Real PreMix (Probe), probe (100 nmol/L), RNase-free ddH_2_O, and 50 × ROX Reference Dye*3. Standard curves were generated by 10-fold serial dilutions (1 × 101 to 1 × 109 copies/μL), and the target group copy number of each reaction was calculated from the standard curves.

### 2.9. Metabolite Concentrations of Colonic Contents

The concentrations of SCFAs (propionic, butyric, and acetic acid) were determined by a gas chromatograph (VARIAN CP-3800, Walnut Creek, CA, USA) with capillary column (30 m × 0.32 mm × 0.25 μm) [29]. The supernatant was mixed with a certain volume of 210 mmol/L crotonic acid and metaphosphoric acid in a new tube after centrifuging (12,000× *g* for 10 min), then those mixtures were centrifuged for 30 min with incubation with identical conditions again at 4 °C. The gas chromatograph was used to analyze 1 μL of the supernatant. The polyethylene glycol column used high-purity N_2_ as carrier gas at a flow rate of 1.8 mL/min.

### 2.10. RNA Isolation, Reverse Transcription and Real-Time Quantitative PCR

Intestinal mucosa (about 100 mg) was homogenized in 1 mL TRIzol reagent, and the total RNA was isolated based on the instructions. Before the RNA samples were reverse transcribed into cDNA with a PrimeScript™ RT reagent kit (Dalian, China), the concentration and fineness of total RNA were analyzed with a spectrophotometer. The qPCR was executed with the SYBR^®^ Green PCR I PCR reagents using the aforesaid uniform PCR Linux, the oligonucleotide primers sequences are shown in Appendix A. The reaction mixture (10 μL) includes the SYBR Premix Ex Taq II, forward and reverse primers, cDNA, and RNase-free ddH_2_O. Cycling in qPCR was as follows: first 95 °C (30 s), then repeat 95 °C (5 s) and 60 °C (30 s) 40 times. β-actin was used as a housekeeping gene to calibrate the mRNA relative expression level of target genes, according to the 2^−ΔΔCt^ method. All the reagents were obtained from Takara Biotechnology Co., Ltd. (Dalian, China).

### 2.11. Statistical Analysis

The data were analyzed by two-way ANOVA with the general linear model (GLM) procedure of SPSS as a 2 (diet) × 2 (breed) factorial design. Differences among the treatments were estimated using a Student–Newman–Keuls multiple comparisons test, and values were indicated as means with their standard errors. The differences were considered significant at *p*-values < 0.05.

## 3. Results

### 3.1. Influence of DF on Growth Performance and Nutrient Digestibility

The Taoyuan pigs had a higher ADFI than the Duroc pigs (Table 2, *p* < 0.05). The ADG was higher in the Taoyuan pigs than in the Duroc pigs under HFD feeding (*p* < 0.05). The F:G ratio between the two pig breeds showed no difference (*p* > 0.05). HFD feeding significantly decreased the apparent digestibilities of DM, GE, and CP in the two pig breeds (*p* < 0.05). The digestibilities of DM, EE, GE, and CP were higher in the Taoyuan pigs than in the Duroc pigs (*p* < 0.05). Moreover, the digestibility of CF was higher in the Taoyuan pigs than in the Duroc under BD feeding (*p* < 0.05). No difference was found in digestibility of ash between the two pig breeds (*p* > 0.05).

### 3.2. Influence of DF on Serum Biochemical Parameters

As shown in Table 3, the serum GSH and T-AOC concentrations were higher in the Taoyuan pigs than in the Duroc pigs (*p* < 0.05). As compared to the Taoyuan pigs, the Duroc pigs had a higher concentration of D-lactate, BUN, and TC in the serum (*p* < 0.05). HFD feeding remarkably reduced the serum MDA concentration, but increased the serum BUN concentration (*p* < 0.05). HFD feeding also improved the serum CAT and T-SOD concentrations in the Taoyuan pigs (*p* < 0.05).

### 3.3. Influence of DF on Intestinal Morphology and Mucosal Enzyme Activity

As shown in Table 4 and Figure 1, the Taoyuan pigs had a higher villus height and ratio of V/C than the Duroc pigs in the jejunum and ileum (*p* < 0.05). HFD feeding remarkably improved the villus height and the V/C ratio in the duodenum and jejunum (*p* < 0.05). The Taoyuan pigs had a lower maltase activity in the jejunal and ileal mucosa than the Duroc pigs (*p* < 0.05). The Duroc pigs also had a higher ileal sucrase activity than the Taoyuan pigs (Table 5). Interestingly, HFD feeding elevated the activities of jejunal maltase and ileal sucrase in the Duroc pigs, and improved activity of ileal sucrase in the Taoyuan pigs (*p* < 0.05).

### 3.4. Influence of DF on Expression Levels of Genes Related to Intestinal Epithelial Functions

As shown in Figure 2, the Taoyuan pigs had a higher expression level of *FATP-1* than the Duroc pigs in the duodenum (*p* < 0.05). HFD feeding not only elevated the expression levels of jejunal and ileal *SGLT-1*, but also increased the ileal *GLUT-2* expression level in the Taoyuan pigs (*p* < 0.05). Additionally, HFD feeding acutely increased the expression levels of jejunal *occludin* and *claudin-1* in both the Taoyuan and Duroc pigs (*p* < 0.05). The duodenal *claudin-1* and ileal *ZO-1* expression levels were also elevated in the two pig breeds upon HFD feeding (*p* < 0.05). In addition, HFD feeding increased the jejunal *ZO-1* expression level in the Duroc pigs (*p* < 0.05).

### 3.5. Influence of DF on Intestinal Microbiota and Microbial Metabolites

As compared to the Duroc pigs, the Taoyuan pigs had a lower *E. coli* abundance and a higher *Bifidobacterium* abundance in the colon (Table 6). However, HFD feeding reduced the colonic *E. coli* abundance in the Duroc pigs (*p* < 0.05). Additionally, HFD feeding remarkably increased the *Lactobacillus* abundance in the Taoyuan pigs (*p* < 0.05). Interestingly, HFD feeding increased the concentrations of acetic acid and propionic acid in the colon (*p* < 0.05). The concentrations of butyrate acid and total acid in the colon were also elevated in the Taoyuan pigs upon HFD feeding (*p* < 0.05).

## 4. Discussion

In this study, we explored the effects of DF on growth performance, nutrient digestibility, and intestinal functions in Taoyuan and Duroc pigs. Pigs were selected based on their ages (60 days of age), as it is difficult to obtain pigs with a similar age and body weight for different pig breeds. We found that the Taoyuan pigs were more resistant to HFD than the Duroc pigs, as indicated by higher ADG and digestibilities of DM, EE, GE, and CP under HFD feeding. As with other local pig breeds, Taoyuan pigs have been reported to digest high-fiber diets more efficiently than commercial pig breeds such as the Duroc and Large White pigs [30,31]. The difference in high-fiber digestion might be associated with the size and microbial activity of the gastrointestinal tract, as the local pig breeds have a greater size (as a percentage of body weight) than that of commercial pig breeds (e.g., Duroc and Large White pigs) [32]. Moreover, adding appropriate DF in the diet increased the abundance and metabolic capacity of distal gut microbiota without altering CF digestibility and growth rate of native pigs [33], which is also consistent with our study. A previous study indicated that HFD feeding may decrease the nutrient digestibility in pigs [34]. In our study, the digestibilities of DM, GE, and CP in the two pig breeds were decreased upon HFD feeding. The decreased nutrient digestibility may be closely associated with the increase in evacuation rate and decrease in the transit time of nutrients due to HFD feeding [26].

Reactive oxygen species (ROS) generated in the body have been implicated in a variety of biological functions such as signal transduction, gene expression, and receptor activation [35]. For instance, ROS signaling was found to regulate cell proliferation, differentiation, migration, immune response, cell senescence and death, and numerous inherited or acquired pathologies such as atherosclerosis, malignant transformation, diabetes mellitus, and aging [36,37,38]. However, overproduction of ROS may conduce disruption of protein conformation and generation of lipid peroxides, leading to damage of their structure and function in cells and tissues [39]. Antioxidative enzymes generated in the body such as T-SOD, GSH, and CAT are responsible for eliminating ROS and play a critical role in maintaining redox homeostasis [40]. A previous study on rats indicated that ingestion of HFD can elevate their antioxidative capacity through modulating the activity of antioxidant enzymes [41]. In the present study, we detect that HFD feeding not just decreased the serum concentration of MDA in the two pig breeds, but also significantly elevated the serum concentrations of CAT and T-SOD in the Taoyuan pigs, which indicated an elevated antioxidative capacity in pigs after HFD ingestion. A previous study indicated that the metabolic differences among pigs with distinct genotypes can be monitored by blood parameters such as the concentration of metabolites and hormones [42]. Urea is produced proportionally to dietary protein levels and has been looked at as an indicator of protein metabolism [43]. In this study, the serum concentrations of BUN were increased in the two pig breeds upon HFD feeding, indicating an increase in protein breakdown (or decrease in protein deposition) [44]. As compared to the Taoyuan pigs, the Duroc pigs had a higher concentration of serum BUN, which suggested that the body protein metabolism may be more sensitive to DF in the Duroc pigs than in the Taoyuan pigs. Moreover, the serum concentration of TG was lower in the Taoyuan pigs than in the Duroc pigs. The result is consistent with a previous report that a local pig breed (e.g., Heigai pig) had lower expression levels of lipolysis-related genes such as the adipose triglyceride lipase (ATGL) and hormone-sensitive lipase (HSL) than commercial pigs (e.g., DLY pig) in the muscle, resulting in a decrease in serum TG concentration [45,46]. Moreover, HFD feeding significantly decreased the serum concentration of glucose in Taoyuan pigs. This is probably due to the reduced substrate (e.g., starch) in the diet that can be enzymatically digested to produce glucose in the small intestine [47,48].

The intestine is the primary location of nutrient digestion and absorption for piglets [49]. Disruption of the villus–crypt integrity (e.g., villus shedding, villus atrophy, and crypt hyperplasia) may lead to invasion of pathogenic bacteria, which subsequently induces various bowel inflammatory diseases [50]. In this study, HFD feeding remarkably increased the villus height and the V/C ratio in the duodenum and jejunum, which suggested an increased absorption area on the surface of the intestinal epithelium [51], and an improved rate of epithelial turnover [52]. The serum D-lactate acid is a critical indicator of intestinal permeability, which is a particular end-product of bacterial fermentation and is released into the blood during disruption of the intestinal mucosa [53]. We found that the serum D-lactate concentration was lower in the Taoyuan pigs than in the Duroc pigs; however, HFD feeding significantly decreased the D-lactate concentration in the Duroc pigs. The result agrees with the intestinal morphology, and both results suggested that local breeds such as the Taoyuan pig may have a better absorption capacity than the Duroc pig. A previous study indicated that changes of the intestinal morphology are followed by alterations in the brush-border enzyme activities [54]. In this study, HFD feeding not only increased the jejunal maltase and ileal sucrase activities in the Duroc pigs, but also increased the ileal sucrase activity in the Taoyuan pigs. Sucrase and maltase are two important disaccharide enzymes that participate in carbohydrate digestion [55]. Moreover, the activities of maltase and sucrase are critical markers to evaluate the development or maturation of the intestinal epithelium [56]. The result is consistent with previous reports that dietary fiber could increase sucrase and maltase in the small intestinal mucosa [57].

The intestinal epithelium protects the host against pathogenic microorganisms through tight junction (TJ) structures [58]. TJ proteins (e.g., *claudin-1* and *ZO-1)* can bind to cytoskeletons, which not only act as staple constituents of the intestinal epithelial epithelium but also act as pivotal regulators of paracellular permeability [59]. A deficit of ZO-1 in mice (specific ZO-1 knockout) showed a disrupted intestinal epithelium, as indicated by abnormal microvillus length and diameter [60]. Moreover, claudin-1 knockdown in mice caused epithelial barrier dysfunction and morphological features of atopic dermatitis in the skin, including hyperkeratosis, acanthosis, and neutrophil infiltration [61]. In the present study, HFD feeding remarkably elevated the expression levels of claudin-1 and occludin in jejunum, and elevated the expression level of ZO-1 in the ileum, which suggested an improved integrity of the intestinal epithelium. Moreover, HFD feeding remarkably improved the expression levels of SGLT-1 and GLUT-2 in the jejunum and ileum of the Taoyuan pigs, and elevated the expression level of jejunal FATP-1 in Duroc pigs. GLUT-2 and SGLT-1 are two of the dominating glucose uptake transporters, and FATP-4 is crucial for long-chain fatty acid absorption in enterocytes [62,63].

DF can be fermented by various microorganisms in the hindgut, and the produced microbial metabolites such as acetic acid, propionic acid, and butyric acid can decrease the lumen pH and inhibit the growth of pathogenic bacteria [64,65,66]. Moreover, the butyric acid can act as a critical energy source of the intestinal epithelial cells [67]. Previous studies also indicated that butyric acid can act as a histone deacetylase (HDAC) inhibitor, which facilitates differentiation of the regulatory T (Treg) cells by up-regulating *Foxp3* expression [68,69]. In the present study, HFD feeding dramatically elevated the concentrations of acetic acid, propionic acid, and butyrate acid in the colon. Propionic acid has also been found to improve the integrity of the intestinal epithelium through promoting the proliferation and differentiation of intestinal epithelial cells and enhancing the expression of tight proteins [70]. Importantly, HFD feeding reduced the abundance of *E. coli* in Duroc pigs and increased the abundance of *Lactobacillus* in Taoyuan pigs. Previous studies indicated that DF can promote the growth of beneficial bacteria such as *Lactobacilli*, and inhibit the growth of several pathogenic bacterial species such as *Escherichia coli* [71,72]. Interestingly, the abundances of *Lactobacilli* and *Bacillus* were higher in Taoyuan pigs than in the Duroc pigs. The result is consistent with previous reports that the intestinal microbial composition is closely related to pig breeds, and local pigs may have a higher abundance of DF-degrading bacteria than commercial pigs (e.g., landrace and Yorkshire pig) [31,73].

## 5. Conclusions

In summary, our results showed a difference in dietary fiber utilization by the two pig breeds (Taoyuan and Duroc), and the results may also suggest a beneficial role of dietary fiber in regulating intestinal health. The Taoyuan pig can digest high-fiber diets more efficiently than commercial pig breeds such as the Duroc pig, and the difference in high-fiber digestion might be connected with the integrity of the intestinal epithelium and the microbial activity of the gastrointestinal tract.

## Figures and Tables

**Figure 1 animals-12-03298-f001:**
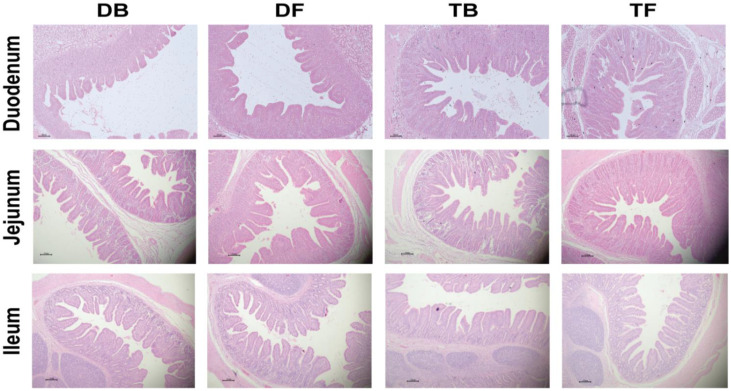
Effect of DF on intestinal morphology in different pig breeds (H&E; × 40). DB, Durocs were exposed to basal diet; DF, Durocs were exposed to high-fiber diet; TB, Taoyuan pigs were exposed to a basal diet; TF, Taoyuan pigs were exposed to a high-fiber diet.

**Figure 2 animals-12-03298-f002:**
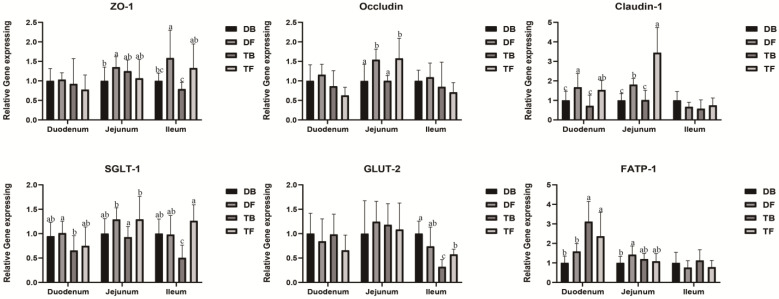
Effect of DF on expressions of critical genes involved in intestinal epithelium functions. ZO-1, zonula occludens-1; SGLT-1, sodium/glucose cotransporter-1; GLUT-2, glucose transporter-2; FATP, fatty acid transport protein-1; ^a^, ^b^, ^c^, mean values within a row with different superscript letters were significantly different (*p* < 0.05). DB, Durocs were exposed to a basal diet; DF, Durocs were exposed to a high-fiber diet; TB, Taoyuan pigs were exposed to a basal diet; TF, Taoyuan pigs were exposed to a high-fiber diet.

**Table 1 animals-12-03298-t001:** Ingredients and nutrient composition of the basal and high-fiber diet *.

Item	Diet	
Ingredients %	BD	HBD
Corn	35.70	35.53
Extruded corn	27.41	27.41
Extruded full-fat soybean	8.49	4.00
Soybean meal	18.00	19.60
Fish meal	4.50	5.90
Soybean oil	1.68	5.20
WBF	0.00	4.30
Sucrose	2.00	0.00
Limestone	0.50	0.30
Dicalcium phosphate	0.58	0.50
NaCl	0.20	0.20
L-Lysine HCl (78%)	0.32	0.40
DL-Methionine	0.09	0.10
L-Threonine (98.5%)	0.02	0.05
L-Tryptophan (98%)	0.01	0.01
Choline chloride	0.15	0.15
Vitamin premix ^1^	0.05	0.05
Mineral premix ^2^	0.30	0.30
Total	100.00	100.00
**Nutrient level (contents)**		
Digestible energy (calculated, MJ/kg)	3.49	3.50
Crude protein	19.15	19.16
Calcium	0.73	0.71
Available phosphorus	0.39	0.40
Lysine	1.22	1.29
Methionine	0.39	0.42
Methionine + cysteine	0.66	0.67
Threonine	0.70	0.72
Tryptophan	0.21	0.33
Soluble dietary fiber	0.68	1.43
Insoluble dietary fiber	2.46	5.43
Total fiber	3.14	6.86

* Based on dry matter. ^1^ Contains vitamins: VA, VD3, VE, VK3, VB2, and VB12; ^2^ minerals: Mn, Cu, I, Zn, Fe, Se, and Ca.

**Table 2 animals-12-03298-t002:** Effects of DF on the performance and nutrient digestibility in different pig breeds.

Item	Duroc		Taoyuan		SEM	*p*-Value		
	DB	DF	TB	TF		Breed	Diet	Interaction
IBW	18.51	18.49	13.88	13.87				
FBW	27.88	26.85	24.29	24.50				
ADFI (g/d)	761.92 ^b^	748.02 ^b^	906.49 ^a^	855.96 ^a^	26.20	0.02	0.52	0.72
ADG (g/d)	374.89 ^ab^	334.22 ^b^	416.60 ^a^	425.40 ^a^	13.18	0.11	0.52	0.32
F:G	2.11	2.06	2.19	2.02	0.04	0.62	0.10	0.59
DM%	83.36 ^b^	80.11 ^d^	86.48 ^a^	81.34 ^c^	0.47	0.01	0.01	0.01
EE%	61.22 ^b^	57.08 ^b^	71.60 ^a^	71.16 ^a^	0.01	0.01	0.18	0.26
GE%	84.36 ^b^	81.20 ^d^	88.12 ^a^	82.78 ^c^	0.01	0.01	0.01	0.01
CP%	77.87 ^b^	73.37 ^d^	83.39 ^a^	75.87 ^c^	0.01	0.01	0.01	0.01
Ash%	41.74	42.91	43.91	42.85	0.01	0.22	0.95	0.21
CF%	25.39 ^b^	46.10 ^a^	56.40 ^a^	52.60 ^a^	2.33	0.01	0.01	0.01

IBW, initial body weight; FBW, final body weight; ADG, average daily gain; ADFI, average daily feed intake; F:G, feed: gain ratio; DM, dry matter; CP, crude protein; EE, ether extract; CF, crude fiber. Mean and total SEM are listed in separate columns, *n* = 10. ^a^, ^b^, ^c^, ^d^ mean values within a row with different superscript letters were significantly different (*p* < 0.05). DB, Durocs were exposed to a basal diet; DF, Durocs were exposed to a high-fiber diet; TB, Taoyuan pigs were exposed to a basal diet; TF, Taoyuan pigs were exposed to a high-fiber diet.

**Table 3 animals-12-03298-t003:** Effects of DF on serum biochemical parameters.

Item	Duroc		Taoyuan		SEM	*p*-Value		
	DB	DF	TB	TF		Breed	Diet	Interaction
BUN (mmol/mL)	5.06 ^b^	6.04 ^a^	2.63 ^d^	4.32 ^c^	0.24	0.01	0.01	0.12
GLU (mmol/mL)	5.07 ^a^	5.30 ^a^	4.92 ^a^	4.26 ^b^	0.11	0.01	0.28	0.03
TC (mmol/mL)	2.02	2.29	2.30	2.34	0.06	0.14	0.15	0.31
TG (mmol/mL)	0.61 ^a^	0.64 ^a^	0.48 ^b^	0.48 ^b^	0.02	0.01	0.76	0.70
DAO (pg/mL)	66.70	73.01	66.00	70.94	1.30	0.59	0.03	0.79
D-lactate (μg/L)	337.02 ^a^	311.93 ^b^	236.75 ^c^	251.79 ^c^	7.40	0.01	0.47	0.01
CAT (U/mL)	33.90 ^a^	31.23 ^a^	25.99 ^b^	29.08 ^a^	1.36	0.07	0.94	0.30
GSH (U/mL)	653.87 ^b^	740.46 ^bc^	779.81 ^ac^	830.89 ^a^	18.38	0.01	0.04	0.58
MDA (U/mL)	2.01 ^a^	1.61 ^b^	2.17 ^a^	1.68 ^b^	0.07	0.03	0.01	0.72
T-AOC (U/mL)	2.43 ^b^	2.45 ^b^	3.42 ^a^	3.40 ^a^	0.18	0.01	0.88	0.82
T-SOD (U/mL)	139.19 ^a^	127.02 ^b^	129.54 ^b^	141.13 ^a^	2.05	0.55	0.94	0.01

CAT, catalase; MDA, malonicdialdehyde; SOD, superoxide dismutase; GSH, glutathione; T-AOC, total antioxidant capacity; TG, triglyceride; TC, total cholesterol; GLU, glucose. Mean and total SEM are listed in separate columns, *n* = 10. ^a^, ^b^, ^c^, ^d^ mean values within a row with different superscript letters were significantly different (*p* < 0.05). DB, Durocs were exposed to a basal diet; DF, Durocs were exposed to a high-fiber diet; TB, Taoyuan pigs were exposed to a basal diet; TF, Taoyuan pigs were exposed to a high-fiber diet.

**Table 4 animals-12-03298-t004:** Effects of DF on intestinal morphology.

Item	Duroc		Taoyuan		SEM	*p*-Value		
	DB	DF	TB	TF		Breed	Diet	Interaction
Duodenum								
Villus height (μm)	338.84 ^b^	430.96 ^a^	347.16 ^b^	441.75 ^a^	10.75	0.54	0.01	0.94
Crypt depth (μm)	347.96 ^a^	325.17 ^ac^	327.19 ^ac^	289.68 ^bc^	8.50	0.09	0.07	0.65
V/C	0.99 ^c^	1.34 ^b^	1.07 ^c^	1.55 ^a^	0.05	0.03	0.01	0.30
Jejunum								
Villus height (μm)	338.43 ^d^	396.13 ^b^	384.35 ^c^	445.85 ^a^	9.71	0.01	0.01	0.90
Crypt depth (μm)	231.83 ^ab^	210.18 ^ab^	232.91 ^a^	208.00 ^b^	4.47	0.95	0.01	0.85
V/C	1.48 ^d^	1.90 ^b^	1.65 ^c^	2.14 ^a^	0.05	0.01	0.01	0.55
Ileum								
Villus height (μm)	330.13 ^b^	361.78 ^ab^	368.55 ^a^	404.37 ^a^	9.30	0.01	0.03	0.86
Crypt depth (μm)	214.90	209.04	211.31	202.57	5.86	0.68	0.55	0.91
V/C	1.51 ^b^	1.79 ^ab^	1.79 ^ab^	2.01 ^a^	0.06	0.02	0.02	0.79

V/C, villus height: crypt depth. Mean and total SEM are listed in separate columns, *n* = 10. ^a^, ^b^, ^c^, ^d^ mean values within a row with different superscript letters were significantly different (*p* < 0.05). DB, Durocs were exposed to basal diet; DF, Durocs were exposed to a high-fiber diet; TB, Taoyuan pigs were exposed to basal diet; TF, Taoyuan pigs were exposed to a high-fiber diet.

**Table 5 animals-12-03298-t005:** Effects of DF on mucosal enzyme activity.

Item	Duroc		Taoyuan		SEM	*p*-Value		
	DB	DF	TB	TF		Breed	Diet	Interaction
Duodenum								
Maltase (U/mgprot)	99.18	98.96	115.76	119.79	5.53	0.11	0.87	0.85
Sucrase (U/mgprot)	19.26	24.14	19.38	21.25	1.61	0.68	0.32	0.65
Lactase (U/mgprot)	5.44	2.54	5.25	5.43	0.55	0.22	0.21	0.16
Jejunum								
Maltase (U/mgprot)	147.72 ^b^	211.42 ^a^	124.97 ^bc^	97.01 ^c^	10.11	0.01	0.22	0.01
Sucrase (U/mgprot)	86.13 ^a^	73.80 ^ac^	61.02 ^c^	61.44 ^c^	3.85	0.15	0.42	0.39
Lactase (U/mgprot)	13.67	16.61	16.23	11.65	0.99	0.54	0.68	0.06
Ileum								
Maltase (U/mgprot)	282.56 ^ac^	350.90 ^a^	179.81 ^b^	243.18 ^bc^	17.78	0.01	0.04	0.94
Sucrase (U/mgprot)	57.13 ^a^	75.01 ^a^	22.41 ^b^	53.90 ^a^	5.28	0.01	0.01	0.45
Lactase (U/mgprot)	2.30	2.66	2.16	2.66	0.17	0.84	0.22	0.85

Mean and total SEM are listed in separate columns, *n* = 10. ^a^, ^b^, ^c^ mean values within a row with different superscript letters were significantly different (*p* < 0.05). DB, Durocs were exposed to basal diet; DF, Durocs were exposed to a high-fiber diet; TB, Taoyuan pigs were exposed to basal diet; TF, Taoyuan pigs were exposed to a high-fiber diet.

**Table 6 animals-12-03298-t006:** Effect of DF on colon microbiota and microbial metabolites in different pig breeds.

Item	Duroc		Taoyuan		SEM	*p*-Value		
	DB	DF	TB	TF		Breed	Diet	Interaction
Microbial populations (lg(copies/g))								
*Escherichia coli*	9.24 ^a^	8.05 ^b^	8.13 ^b^	8.11 ^b^	0.15	0.07	<0.05	0.04
*Lactobacillus*	7.68 ^b^	7.68 ^b^	7.71 ^b^	8.32 ^a^	0.10	0.10	0.12	0.12
*Bifidobacterium*	6.37 ^b^	6.12 ^b^	6.80 ^a^	6.81 ^a^	0.08	0.01	0.39	0.31
*Bacillus*	8.92	8.93	8.94	8.99	0.04	0.56	0.65	0.79
Total bacteria	12.01 ^ab^	12.05 ^a^	11.89 ^b^	12.01 ^ab^	0.03	0.15	0.13	0.47
VFA (g/g)								
Acetic acid	2.90 ^b^	3.49 ^a^	2.78 ^b^	3.57 ^a^	0.11	0.92	0.01	0.60
Propionic acid	1.26 ^b^	1.58 ^a^	1.17 ^b^	1.54 ^a^	0.05	0.45	0.01	0.82
Butyrate acid	0.62 ^ab^	0.72 ^ab^	0.55 ^b^	0.83 ^a^	0.04	0.79	0.02	0.23
Total acid	5.24 ^b^	6.16 ^ab^	5.07 ^b^	6.28 ^a^	0.18	0.94	0.01	0.67

Mean and total SEM are listed in separate columns, *n* = 10. ^a^, ^b^, ^c^ mean values within a row with different superscript letters were significantly different (*p* < 0.05). DB, Durocs were exposed to a basal diet; DF, Durocs were exposed to a high-fiber diet; TB, Taoyuan pigs were exposed to a basal diet; TF, Taoyuan pigs were exposed to a high-fiber diet.

## Data Availability

The data used to support the findings of this study are available from the corresponding author upon request.

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
