# Peer review of "Effects of Dietary Fiber on Growth Performance, Nutrient Digestibility and Intestinal Health in Different Pig Breeds"

_animals, 2022, doi:10.3390/ani12233298_

Round 1

Reviewer 1 Report

Dietary fiber has been a hot area in human and animal nutrition. The research paper provide information to better understand the different response to the dietary fiber of different animals, it has a decent value in helping producers come up with proper strategies in production. Although the language and writing of the manuscript are of good quality, some key improvements must be made.

To start with, dietary fiber is the sum of soluble and non-soluble fiber from plant-based foods which contains pectins, gums, and mucilage, while crude fiber is the insoluble part of the cell wall of plants which is mostly consisted of cellulose (and also lignin in woody plants). It looks like the author mixed up the two concepts in preparing the manuscript, they are not interchangeable measurements. This needs to be fixed including:

1)      Line 81-82, explain why dietary fiber was switched to crude fiber? Dietary fiber content numbers are needed, not crude fiber;

2)      Line 87, for the dietary fiber research, you need to provide the dietary fiber content of both dietary treatment and formulas of both diets;

3)       Line 194, for digestibility results, dietary fiber, and crude fiber need to be evaluated.

Secondly, the design of the study needs to be better described including:

1)       Line 79, why pigs of different weights were selected for this study, and are they all 60 days old pigs? Is there any variance in age?

2)      Line 194, the Result table needs to have body weight information to avoid misleading the readers.

3)      Line 241, the discussion part needs to explain the comparable results of the two breeds at such a different body weight.

Lastly, it looks like high dietary fiber treatment improved the growth performance of the Taoyuan breed pigs although it is not significant. Also, why are there multiple comparisons when there is no significant result detected? The decrease in dry matter digestibility in both breeds is contradicted by the improved growth performance and reduced feed intake. These results need to be re-evaluated, are there any outliers for growth performance data?   

Author Response

Point to point response to reviewers comments

Thank you very much for giving us an opportunity to submit a revised version of our manuscript. We appreciate for your comments and suggestions concerning our manuscript. These comments are valuable and helpful for revising and improving our manuscript, and we revised the manuscript in accordance with the detailed comments and suggestions. All revisions are highlighted in red in the text. The point-by-point revisions to the comments and suggestions are listed as follows:

Reviewer #1:

Comments and Suggestions for Authors

To start with, dietary fiber is the sum of soluble and non-soluble fiber from plant-based foods which contains pectins, gums, and mucilage, while crude fiber is the insoluble part of the cell wall of plants which is mostly consisted of cellulose (and also lignin in woody plants). It looks like the author mixed up the two concepts in preparing the manuscript, they are not interchangeable measurements. This needs to be fixed including

  • Line 81-82, explain why dietary fiber was switched to crude fiber? Dietary fiber contentnumbers are needed, not crude fiber;

Re: Thanks for your comments. We are sorry for the confused description of the “dietary fiber” and “crude fiber”. It has been corrected in the revised manuscript. Moreover, the component of dietary fiber (soluble and insoluble content) has been added in the article (Table 1). 

  • Line 87, for the dietary fiber research, you need to provide the dietary fiber content of both dietary treatment and formulas of both diets;

Re: Thanks for your comments. We have added the dietary fiber content of both dietary treatment and formulas of both diets in the revised manuscript.

  • Line 194, for digestibility results, dietary fiber, and crude fiber need to be evaluated.

Re: Thanks for your comments. The digestibility of crude fiber has been added in the revised manuscript.

Secondly, the design of the study needs to be better described including:

  • Line 79, why pigs of different weights were selected for this study, and are they all 60 days old pigs? Is there any variance in age?

Re: Thanks for your comments. The difference in body weight results from their breeds. It is impossible to select these pigs with a similar body weight and developmental age. In this study, all the pigs are 60 days old. This has been addressed in the revised manuscript.

  • Line 194, the Result table needs to have body weight informationto avoid misleading the readers.

Re: Thanks for your comments. We have added the body weight information in the revised manuscript.

  • Line 241, the discussion part needs to explain the comparable results of the two breeds at such a different body weight.

Re: Thanks for your comments. We already have added discussion to explain the comparable results of the two breeds such as different body weight in the revised manuscript.

  • Lastly, it looks like high dietary fiber treatment improved the growth performance of the Taoyuan breed pigs although it is not significant. Also, why are there multiple comparisons when there is no significant result detected? The decrease in dry matter digestibility in both breeds is contradicted by the improved growth performance and reduced feed intake. These results need to be re-evaluated, are there any outliers for growth performance data?   

Re: Thanks for your comments. We gave checked and re-analyzed the growth performance data. In this study, the growth performance (e.g. ADG) of the Taoyuan pig was not affected by dietary fiber (416 vs. 425 g/d). However, the ADFI was significantly affected by the pig breeds. The multiple comparisons offer more information about the genetic variation. Actually, theres a tendency to decrease the growth performance (ADG) in Duroc pig (334 vs. 374 g/d) upon high-fiber diet feeding (0.05<P<010). These results are consistent with previous reports that local pig breeds can digest dietary fibers more efficiently than other commercial pig breeds [1-3]. This has also been discussed in the revised manuscript.

[1] Yang, L.; Bian, G.; Su, Y. Comparison of faecal microbial community of lantang, bama, erhualian, meishan, xiaomeishan, duroc, landrace, and yorkshire sows. Asian-Australasian journal of animal sciences, 2014, 27(6), 898–906.

[2] Martins, J.M.; Fialho, R.; Albuquerque, A. Growth, blood, carcass and meat quality traits from local pig breeds and their crosses. Animal : an international journal of animal bioscience, 2020, 14(3), 636–647.

[3] Borin K, Lindberg J E, Ogle R B. Effect of variety and preservation method of cassava leaves on diet digestibility by indigenous and improved pigs. Animal Science, 2005, 80(3), 319–324.

Reviewer 2 Report

Dear Authors, the paper entitled Effects of Dietary Fiber on Growth Performance, Nutrient Digestibility and Intestinal Health in Different Pig Breeds, is interesting, and presents some good results. Although in my opinion the paper is not exactly on the topic of the Foods journal, it fits perfect with the topic of the Special Issue "Carbohydrates and Intestinal Health"

Please find my observations below:

Obs 1. What was the age of the pigs? It is very clear indeed that the experiment lasted 28 days, but in what interval? Weaning, post weaning, fattening … Please mention this aspect in the abstract. Also, the two breeds of pigs used are pure breeds?

Row 84, in a metabolic cage.

Row 87, Table 1, delete the % after corn

Row 91 and 94 please replace gathered with collected.

Row 96 - up to text? What the authors wanted to say?

Row 96. Behind? Or After the blood sampling …

Row 104. Same observation  The morning behind the experiment… I think is after …

Row 194, Table 2,  for ADFI , reverse the letters of significance among the groups.

Figure 1 and table 4 should be moved at row 214 3.3. Influence of DF on intestinal morphology and mucosal enzyme activity, the results should be presented at the results chapter not in discussions.

In Table 4 has the same problem as table 2.  The letter a, which indicates the significance should be places at the highest value, after b, c and so on. In this table the letters are placed like in chop soup of letters. For example, the Villus height in duodenum, the letter a was placed at the lowest values, after for duodenum Crypt depth, the letter a was placed for the highest value. Please double check and uniform them to have sense and easy reading and understanding of the reported results. Under the Table as a footnote please add all the letters a, b, c, d mean

Effects of DF on mucosal enzyme activity is in fact Table 5. This table with results needs to be moved in the results section at row 214 3.3. Influence of DF on intestinal morphology and mucosal enzyme activity

Moreover, the surcease in the ileum, of TB group, are the authors sure about those differences?  Having a SEM of 5.28 and the P value at 0.01, I really think that DF is significantly higher than DB and TF, and these two, higher than TB. Please check again these values.

Figure 2, should be moved at 223 section 3.4. Influence of DF on the mRNA level of genes related to intestinal epithelial functions

Table 6. Effect of DF on intestinal microbiota and microbial metabolites in different pig breeds must be moved in the results section at row 233 section 3.5. Influence of DF on intestinal microbiota and microbial metabolites

 In the discussion section, the results presented in Table 3. Effects of DF on serum biochemical parameters, are not enough discussed and compared. This part needs improvements.

The gut health part is also very poor discussed. The authors can add some information to enlarge it.

Lastly, the language needs some revision. Please carefully revise your manuscript and improve it's quality of presentation. 

Author Response

Point to point response to reviewer’s comments

Thank you very much for giving us an opportunity to submit a revised version of our manuscript. We appreciate for your comments and suggestions concerning our manuscript. These comments are valuable and helpful for revising and improving our manuscript, and we revised the manuscript in accordance with the detailed comments and suggestions. All revisions are highlighted in red in the text. The point-by-point revisions to the comments and suggestions are listed as follows:

Reviewer #1:

Comments and Suggestions for Authors

Obs 1. What was the age of the pigs? It is very clear indeed that the experiment lasted 28 days, but in what interval? Weaning, post weaning, fattening … Please mention this aspect in the abstract. Also, the two breeds of pigs used are pure breeds?

Re: Thanks for your comments. We are sorry for the missing of some detailed information for the pigs. The pigs utilized in this study were 60 d after birth, and the trial lasted for 28 days. Moreover, the two breeds of pigs were pure breeds. These details have been added in the revised manuscript.

Row 84, in a metabolic cage.

Re: Thanks for your comments. It has been corrected in the revised manuscript.

Row 87, Table 1, delete the % after corn

Re: Thanks for your comments. It has been deleted in the revised manuscript.

Row 91 and 94 please replace gathered with collected.

Re: Thanks for your comments. It has been corrected in the revised manuscript.

Row 96 - up to text? What the authors wanted to say?

Re: Thanks for your comments. We are sorry for the confused description. This sentence has been corrected in the revised manuscript.

Row 96. Behind? Or After the blood sampling …

Re: Thanks for your comments. The word “Behind” has been replaced by “After” in the revised manuscript.

Row 104. Same observation  The morning behind the experiment… I think is after …

Re: Thanks for your comments. It has been corrected in the revised manuscript.

Row 194, Table 2,  for ADFI , reverse the letters of significance among the groups.

Re: Thanks for your comments. It has been corrected in the revised manuscript.

Figure 1 and table 4 should be moved at row 214 3.3. Influence of DF on intestinal morphology and mucosal enzyme activity, the results should be presented at the results chapter not in discussions.

Re: Thanks for your comments. We have moved Figure 1 and table 4 at row 214. “3.3. And the results of influence of DF on intestinal morphology and mucosal enzyme activity” have been moved at the results chapter.

In Table 4 has the same problem as table 2. The letter a, which indicates the significance should be places at the highest value, after b, c and so on. In this table the letters are placed like in chop soup of letters. For example, the Villus height in duodenum, the letter a was placed at the lowest values, after for duodenum Crypt depth, the letter a was placed for the highest value. Please double check and uniform them to have sense and easy reading and understanding of the reported results. Under the Table as a footnote please add all the letters a, b, c, d mean

Re: Thanks for your comments. We are sorry for this mistake. It has been corrected in the revised manuscript.

Effects of DF on mucosal enzyme activity is in fact Table 5. This table with results needs to be moved in the results section at row 214 3.3. Influence of DF on intestinal morphology and mucosal enzyme activity

Re: Thanks for your comments. This table with results have been moved in the results section at row 214 3.3. Influence of DF on intestinal morphology and mucosal enzyme activity.

Moreover, the surcease in the ileum, of TB group, are the authors sure about those differences?  Having a SEM of 5.28 and the P value at 0.01, I really think that DF is significantly higher than DB and TF, and these two, higher than TB. Please check again these values.

Re: Thanks for your comments. We have carefully checked the manuscript and there is no problem. Actually, for most parameters, the SEM is acceptable.

Figure 2, should be moved at 223 section 3.4. Influence of DF on the mRNA level of genes related to intestinal epithelial functions

Re: Thanks for your comments. Figure 2 has been moved at 223 section 3.4. Influence of DF on the mRNA level of genes related to intestinal epithelial functions.

Table 6. Effect of DF on intestinal microbiota and microbial metabolites in different pig breeds must be moved in the results section at row 233 section 3.5. Influence of DF on intestinal microbiota and microbial metabolites

Re: Thanks for your comments. “Table 6. Effect of DF on intestinal microbiota and microbial metabolites in different pig breeds” has been moved in the results section at row 233 section 3.5. Influence of DF on intestinal microbiota and microbial metabolites.

 In the discussion section, the results presented in Table 3. Effects of DF on serum biochemical parameters, are not enough discussed and compared. This part needs improvements.

Re: Thanks for your comments. We have added more discussion and comparison about the serum biochemical parameters in the revised manuscript.

The gut health part is also very poor discussed. The authors can add some information to enlarge it.

Re: Thanks for your comments. We have paid more attentions on the discussion of the gut heath in the revised manuscript.

Lastly, the language needs some revision. Please carefully revise your manuscript and improve it's quality of presentation.

Re: Thank you for your suggestion. The English writing and style in this manuscript have been carefully checked and corrected.

Round 2

Reviewer 2 Report

The authors corrected and improved the paper according to the previous comments. Now it can be accepted for publication.

As a last observation please be aware that the statistical analyses in Table 2 for BW are missing.

Goog luck!